# The Role of AKR1B10 in Lung Cancer Malignancy Induced by Sublethal Doses of Chemotherapeutic Drugs

**DOI:** 10.3390/cancers16132428

**Published:** 2024-07-01

**Authors:** Te-Hsuan Jang, Sheng-Chieh Lin, Ya-Yu Yang, Jong-Ding Lay, Chih-Ling Chang, Chih-Jung Yao, Jhy-Shrian Huang, Shuang-En Chuang

**Affiliations:** 1National Institute of Cancer Research, National Health Research Institutes, Miaoli 35053, Taiwan; jang@nhri.edu.tw (T.-H.J.); jaysclin@gmail.com (S.-C.L.); irelandfish@nhri.edu.tw (Y.-Y.Y.); forrest70312@gmail.com (C.-L.C.); 2Department of Nursing, National Taichung University of Science and Technology, Taichung 40343, Taiwan; jdlay@nutc.edu.tw; 3Department of Internal Medicine, School of Medicine, College of Medicine, Taipei Medical University, Taipei 11031, Taiwan; yao0928@tmu.edu.tw; 4Department of Medical Education and Research, Wan Fang Hospital, Taipei Medical University, Taipei 11031, Taiwan; 5Division of Endocrinology and Metabolism, Department of Internal Medicine, Ditmanson Medical Foundation Chiayi Christian Hospital, Chiayi City 60002, Taiwan

**Keywords:** sublethal doses, non-small-cell lung cancer, AKR1B10

## Abstract

**Simple Summary:**

The present research investigates the counterintuitive effects of sublethal chemotherapy doses in non-small-cell lung cancer (NSCLC), focusing on their role in enhancing cancer cell malignancy and the role of Aldo-keto reductase family 1 member B10 (AKR1B10) in this process. This study is designed to understand how sublethal doses of chemotherapy drugs like taxol and doxorubicin lead to increased cancer cell migration, invasion, metastasis, and the consequent upregulation of AKR1B10. Our findings uncover the fact that AKR1B10 plays a crucial role in NSCLC progression and chemoresistance, with its increased expression being linked to enhanced malignancy. The results are significant for the research community as they provide insight into the currently paradoxical molecular mechanisms of chemotherapy-induced resistance in lung cancer. This could lead to the potential development of more effective treatment strategies that consider the impact of chemotherapy dosing on cancer cell behavior, potentially improving patient outcomes by preventing the unintentional enhancement of cancer aggressiveness.

**Abstract:**

Chemotherapy remains a cornerstone in lung cancer treatment, yet emerging evidence suggests that sublethal low doses may inadvertently enhance the malignancy. This study investigates the paradoxical effects of sublethal low-dose chemotherapy on non-small-cell lung cancer (NSCLC) cells, emphasizing the role of Aldo-keto reductase family 1 member B10 (AKR1B10). We found that sublethal doses of chemotherapy unexpectedly increased cancer cell migration approximately 2-fold and invasion approximately threefold, potentially promoting metastasis. Our analysis revealed a significant upregulation of AKR1B10 in response to taxol and doxorubicin treatment, correlating with poor survival rates in lung cancer patients. Furthermore, silencing AKR1B10 resulted in a 1–2-fold reduction in cell proliferation and a 2–3-fold reduction in colony formation and migration while increasing chemotherapy sensitivity. In contrast, the overexpression of AKR1B10 stimulated growth rate by approximately 2-fold via ERK pathway activation, underscoring its potential as a target for therapeutic intervention. The reversal of these effects upon the application of an ERK-specific inhibitor further validates the significance of the ERK pathway in AKR1B10-mediated chemoresistance. In conclusion, our findings significantly contribute to the understanding of chemotherapy-induced adaptations in lung cancer cells. The elevated AKR1B10 expression following sublethal chemotherapy presents a novel molecular mechanism contributing to the development of chemoresistance. It highlights the need for strategic approaches in chemotherapy administration to circumvent the inadvertent enhancement of cancer aggressiveness. This study positions AKR1B10 as a potential therapeutic target, offering a new avenue to improve lung cancer treatment outcomes by mitigating the adverse effects of sublethal chemotherapy.

## 1. Introduction

Adenocarcinoma (ADC) and squamous cell carcinoma are the two major subtypes of non-small-cell lung cancer (NSCLC) that account for at least 85% of lung cancers [1]. Despite the fact that recent advances in therapeutic modalities for NSCLC have largely improved treatment response, the 5-year overall survival (OS) remains poor at 28% [2,3]. Chemotherapy has been recognized as the mainstay therapeutic option for all stages of NSCLC patients for prolonging survival after surgery [4]. Regardless of the initial responsiveness of tumors to chemotherapies, lung cancer patients, especially those in advanced stages, rapidly develop chemoresistance leading to disease recurrence and progression [1,5]. Over the years, enormous efforts toward understanding the mechanisms that underline chemoresistance have been made, revealing the involvements of elevated drug metabolism, genetic alterations, and epigenetic alterations, as well as cellular adaptations, to mitigate the insults from chemotherapeutic agents [6]. Nonetheless, limited treatment options are available following the acquisition of resistance, making resistance to conventional chemotherapy a major clinical hurdle that impairs effective cancer treatment [7,8]. It is thus crucial that new approaches based on inadvertent molecular mechanisms are identified to alleviate cellular resilience in refractory tumors.

Because most of our current knowledge on chemotherapeutic resistance emerges from comparing drug-resistant with drug-sensitive cells or tumors, little has been explored in regard to the temporal development of chemoresistance. To date, multidisciplinary approaches that target drug resistance mechanisms such as dysregulated pathways of DNA repair, pro-survival signaling, drug efflux, cell cycle arrest, microRNA, and cancer metabolism have been proposed to delay the development of therapeutic resistance or to re-sensitize lung cancers [5,9,10]. Of note, recent studies have suggested that the drug uptake of tumors is pertinently attributed to their distances from blood vessels [11,12], implicating the importance of spatiotemporal delivery for effective drug concentrations to the sites of cancer cells [13]. Sublethal doses of drugs are renowned for the subsequent development of cellular resistance and have been utilized for establishing chemoresistant study models from the surviving cancer cells [14,15,16].

AKR1B10 is a member of the aldo–keto reductase (AKR) superfamily that was identified as a novel protein correlated to the carcinogenesis of hepatocellular carcinoma (HCC), albeit the fact that the primary role of AKR1B10 is protecting gastrointestinal cells against carbonyl damage [17,18]. Although AKR1B10 is downregulated in gastrointestinal cancers and inflammatory bowel diseases [19,20], it is later found to be differentially upregulated and proposed as a prognostic biomarker in numbers of other cancer types, such as nasopharyngeal carcinoma, breast, pancreatic, and lung cancers [21,22,23]. In addition, AKR1B10 is shown to be greatly elevated in the brain metastasis of lung cancer and the expression in the serum of lung cancer patients with bone metastasis, which serves as a novel diagnostic biomarker [24]. Moreover, AKR1B10 inhibitors significantly inhibit the metastasis and invasion abilities of cisplatin-resistant lung cancer cells [25]. Therefore, accumulating evidence supports AKR1B10 as a potential target for cancer diagnosis and treatment.

In this study, we first employed escalating but sublethal doses of four FDA-approved chemodrugs to NSCLC cells with the aim of determining the most significantly up-regulated genes during the temporal chemoresistance development using microarray analysis. The gradated administration of NSCLC cells to chemodrugs unveiled Aldo-keto reductase family 1 member B 10 (AKR1B10) as the most potent regulator in the resilience development that was functionally characterized in the present study.

## 2. Materials and Methods

### 2.1. Cell Lines, Transfection and Chemical Reagents

Three human NSCLC cell lines (CL1-0, CL1-3 and PE089) were used in this study [26,27,28]. CL1-0 and CL1-3 cell lines were cultured in an RPMI-1640 medium, while PE089 cells were cultured in an MEM medium, each supplemented with 10% fetal bovine serum (FBS) and a 1% penicillin/streptomycin. The cells were incubated at 37 °C in a humidified chamber with 5% CO_2_. Transfection was carried out at 70% cell confluence following the manufacturer’s instructions, utilizing a Lipofectamine 2000 transfection reagent (Invitrogen, Carlsbad, CA, USA). After transfecting the AKR1B10 plasmids into the cells for 48 h, the cell lysates were collected for Western blot analysis and AKR1B10 activity assay. Cisplatin was acquired from Abic Biological Laboratories (Yerushalayim, Israel), doxorubicin from Sigma–Aldrich (St. Louis, MO, USA) (Sigma D-1515), gemcitabine from Eli Lilly (Gemzar, Eli Lilly and Company, Indianapolis, IN, USA), and taxol from the Yung Shin Pharmaceutical Industrial Co., Ltd. (Taichung, Taiwan).

### 2.2. Cell Viability Assay and Clonogenic Survival Assay

To determine the cell viability of CL1-3 cells exposed various anticancer drugs, we performed a tetrazolium-based semi-automated colorimetric method using MTT assay (Roche, Indianapolis, IN, USA) according to the manufacturer’s protocol. Briefly, CL1-3 cells were seeded (2000 cells/well) into a 96-well plate. After overnight culture, the cells were exposed to the indicated doses of anticancer drugs for 3 days. With sublethal doses, the cell mortality rates were maintained at approximately 50% during the drug administration, which allowed for the remaining cells to survive and recover. The cell viability was evaluated with an ELISA reader at O.D. 540 nm. For colony formation assays, the cells were incubated with the indicated concentrations of anticancer drugs for 3 days were trypsinized and seeded (400 cells/well) into a 6-well plate until they were stained by crystal violet after 11 days. The number of colonies containing more than 100 cells was counted. The results are presented as the mean ± standard deviation (SD) of three independent experiments.

To determine the potential functions of the AKR1B10 involved in cell proliferation, we transfected CL1-3 or PE089 cells with AKR1B10-specific siRNA and scrambled siRNA, which were procured from Dharmacon (Lafayette, CO, USA). After transfection, the cells were trypsinized and seeded (2 × 10^3^ cells/well) into a 96-well plate (400 cells/well) or into a 6-well plate. Then, these cells were exposed to various doses of anticancer drugs for 11 days before the colony formation assays.

### 2.3. Cell Migration Assay

We utilized culture inserts (Cat. No. 80209, ibidi™) sourced from Thistle Scientific Ltd. (Cheshire, UK). The CL1-3 or PE089 cells treated with chemotherapy were prepared at a concentration of 6 × 10^5^ cells/mL in the medium, and 70 μL of this cell suspension was added to each silicon insert. These inserts were subsequently removed once the cells reached approximately 100% confluence. To quantify the migration, we captured photographs at two distinct time points following the removal of the silicon inserts and measured the migration distance.

### 2.4. Matrigel Invasion Assay

The in vitro invasion assays were conducted using the BD Biocoat Matrigel invasion chamber system from BD Biosciences (Bedford, MA, USA). Initially, 1 × 10^4^ cells were seeded onto the upper chamber of the Matrigel invasion chamber and incubated at 37 °C for 24 h. Following incubation, the invasive cells were collected. Those cells that had successfully penetrated the membrane were then fixed in methanol, stained with Giemsa stain and subsequently counted under a microscope.

### 2.5. Microarray Analysis

The CL1-3 cells underwent a pre-treatment phase with or without non-toxic and sublethal doses of doxorubicin (60 nM) and taxol (4.5 nM), as indicated, for a duration of 3 days. Following this treatment, the cells were thoroughly washed to remove the drugs and subsequently cultured in a drug-free medium for two passages. To perform the gene expression microarray analysis on cells treated with these chemotherapeutic drugs, the total RNA was extracted using Trizol reagent from Invitrogen (Carlsbad, CA, USA). The RNA samples were then utilized for the analysis, which was conducted on the Affymetrix HG U133 Plus 2.0 arrays (comprising 54,675 probe sets, Affymetrix, Santa Clara, CA, USA). This analysis was carried out at the core facility of NHRI, as previously described. To ensure consistency and comparability, the signal values for all genes were log-transformed to the base 2, and quantile normalization was applied to achieve uniform distributions of the probe signal intensities.

### 2.6. AKR1B10 Plasmid Construction

The AKR1B10 cDNA was synthesized through reverse-transcription-PCR (RT-PCR). The AKR1B10 cDNA was PCR-amplified using sense primer (5′-GGAATTCATGGCCACGTTTGTGG-3′) and antisense primer (5′-GCTCGAGTCAATATTCTGCATTG-3′) and cloned into the *EcoR*I and *Xho*I sites of the pcDNA3 vector. The resulting AKR1B10 cDNA expression construct was confirmed by DNA sequencing.

### 2.7. AKR1B10 Activity Assay

Reductase activity in the cell extracts was measured as a decrease in nicotinic acid adenine dinucleotide phosphate (NADPH) absorbance at 340 nm. [29]. Briefly, the cells were lysed on ice in a lysis buffer containing NaH_2_PO_4_ (20 mM; pH 7.0), β-mercaptoethanol (2 mM), leupeptin (5 μM), and phenylmethylsulfonyl fluoride (20 μM) for 30 min, followed by centrifugation at 10,000× *g* and 4 °C for 10 min. The soluble proteins (50 μg) were used for the AKR1B10 activity assay in a reaction mixture consisting of 125 mM sodium phosphate (pH 7.0), 0.2 mM reduced nicotinamide adenine dinucleotide phosphate (NADPH), 50 mM KCl, and 20 mM DL-glyceraldehyde. After incubation at 35 °C for 10 min, the oxidized NADPH was monitored at 340 nm for AKR1B10 activity. The expression and functionality of the AKR1B10 protein in the CL1-0 and CL1-3 cells were verified by Western blot and enzymatic activity, respectively.

### 2.8. Data Mining of Clinical Information on AKR1B10

The RNA expression profile of AKR1B10 was downloaded from the GPL570 platform datasets through GENT2 (http://gent2.appex.kr/gent2/, accessed on 2 September 2023) and the GDC TCGA LUAD dataset. The interactive body map for the expression of AKR1B10 in normal and tumorous lungs was generated by Gene Expression Profiling Interactive Analysis 2 (GEPIA2; http://gepia2.cancer-pku.cn/#index, accessed on 12 September 2023), based on both the TCGA and the GTEx databases. The survival analysis was performed by using lung adenocarcinoma pan-cancer RNAseq data from the Kaplan–Meier (KM) plotter (https://www.kmplot.com/analysis/, accessed on 12 September 2023). Patients with high or low AKR1B10 expression were divided based on the optimal cutoff value determined by the KM plotter.

### 2.9. Western Blot Analysis

The cells were lysed on ice in a radioimmunoprecipitation assay buffer (RIPA buffer) containing protease inhibitors (Roche, IN, USA). The protein concentrations were measured by using the BCA protein assay kit (ThermoFisher Scientific Biosciences, Rockford, IL, USA). The quantified protein lysates were separated by SDS-PAGE and transferred onto a PVDF membrane (Millipore, Billerica, MA, USA). The membranes were incubated with an antibody against AKR1B10, cyclin A, cyclin D1, cyclin E, ERK, phosphorylated ERK, actin, or tubulin at 4 °C overnight. After incubation with an HRP-conjugated secondary antibody, the immunoreactive bands were visualized by an enhanced chemiluminescence detection system (Millipore). The protein bands were quantified by ImageJ 1.54i 4.0 software.

### 2.10. In Vivo Metastases Assay

Male NOD/SCID mice 4–6 weeks old were obtained from BioLASCO (Taipei, Taiwan). The animal study protocols were approved by the Institutional Animal Care and Use Committee of NHRI. For Model I, the CL1-3 cells were pre-treated with non-toxic and sublethal low doses of chemotherapy drugs as indicated for 3 days, then the drugs were washed out and the cells were cultured in a drug-free medium for two passages. Then, the treated cells were intravenously injected into mice via the tail vein. Six weeks later, the mice were sacrificed and examined for lung metastases.

### 2.11. Statistical Analysis

Quantitative data were presented as mean ± SD. The differences between the control and treatment groups were analyzed by Student’s *t*-test. A probability level of *p* < 0.05 was considered statistically significant in each experiment.

## 3. Results

### 3.1. Establishing Sublethal Doses of Chemotherapy Drugs for Lung Adenocarcinoma Cells

To determine the appropriate sublethal doses of chemotherapeutics on CL1-3 lung adenocarcinoma cell lines that represent Taiwanese lung cancer patients, taxol, doxorubicin, cisplatin, and gemcitabine were administered across a broad range of concentrations to those cells. The resulting cytotoxic effects exerted circumscribed doses of all four drugs utilized to evaluate their long-term, enduring cytotoxic effects on cell survival for the colony formation assay. The sublethal doses maintained the cell mortality rate at approximately 50% during the drug administration, which allowed for the remaining cells to survive. Data derived from the colony formation assay revealed that the sublethal doses for taxol (3 and 4.5 nM), doxorubicin (40 and 60 nM), cisplatin (1 and 2.5 μM), and gemcitabine (4 and 8 nM) effectively compromised the colony formation ability of the CL1-3 cells (Figure 1A–D).

### 3.2. Escalating Sublethal Doses of Chemotherapeutics Elevates Migration and Invasion Ability of Lung Adenocarcinoma Cells

The same range of doses for all four chemotherapeutics were next applied and assessed for their influences on the cellular migration and invasion abilities. Notably, the cellular migration ability, as indicated by wound healing rates, was significantly enhanced when the CL1-3 cells were treated with higher sublethal doses, except for doxorubicin, which at 5 nM was adequate to significantly increase the migration ability as compared to the vehicle control-treated cells (Figure 2A). Further, the cellular invasion ability of the CL1-3 cells was also significantly augmented by all four drugs at lower sublethal doses from as low as 0.3 nM of taxol (Figure 2B).

### 3.3. NSCLC In Vivo Lung Metastasis Is Augmented upon Sublethal Dose Treatments

The CL1-3 lung cancer cells that received the sublethal chemotherapeutic treatments were passaged two times to allow recovery from the insults prior to the intravenous injection into mice for lung metastasis assessments (Table 1). The results showed that the number of metastasized lung nodules from the anticancer drug-treated groups were all significantly greater than that of the PBS-treated group, implying that the sublethal doses utilized in the present study were sufficient to induce the metastasis of the CL1-3 cells.

### 3.4. Expression and Activity of AKR1B10 Is Upregulated in Response to Sublethal Chemotherapeutic Regimens

Our microarray analysis unveiled AKR1B10 as the top differentially upregulated gene in response to sublethal chemotherapeutic insults from taxol and doxorubicin in NSCLC (Figure 3A). The cellular AKR1B10 activity in the CL1-3 cells was also found to be significantly enhanced in chemo-insulted cells at two passages as compared to PBS-treated cells (Figure 3B). We evaluated the clinical significance of AKR1B10 using the GPL570 platform dataset (Figure 3C), a microarray platform developed by Affymetrix for gene expression profiling commonly referred to as the Affymetrix Human Genome U133 Plus 2.0 Array, and the GDC TCGA LUAD dataset (Figure 3D), a comprehensive genome, and clinical data of lung adenocarcinoma (LUAD) collected and made available by the Cancer Genome Atlas (TCGA) through the Genomic Data Commons (GDC). We found that the increased AKR1B10 mRNA expression was strongly associated with lung tumor tissue as compared to normal tissue. We also used GEPIA2 to look at ARK1B10 mRNA expression in human normal tissue (left, green) or tumor tissue (right, red) showing specific levels in the lung (arrow; normal lung: 0.2; tumor lung: 68.23) (Figure 3E). The subsequent Kaplan–Meier (KM) plotter survival analysis of 504 lung adenocarcinoma patients revealed an association between higher ARK1B10 levels and poorer overall survival (Figure 3F)

### 3.5. AKR1B10 Silencing Reduces Cell Proliferation, Migration, and Invasion in NSCLC

The biological function of AKR1B10 in response to chemo-insults was first investigated by examining the cellular proliferation rate of CL1-3 and PE089 cells that were transfected with AKR1B10 siRNA. Figure 4A demonstrates the siRNA that specifically targeted AKR1B10 resulted in a significantly lowered growth rate in both the CL1-3 and PE089 cells as compared to the scramble-control (sc) transfected counterparts. The cell cycle regulators, such as cyclins, are known for their role in the cellular proliferation of NSCLC [30]; we hence evaluated whether major cyclins, including cyclin A, cyclin D, and cyclin E, were involved in the AKR1B10-mediated proliferation in the CL1-3 and PE089 cells. Our results show that when AKR1B10 was downregulated by siAKR, the expressions of cyclin A, cyclin D and cyclin E were markedly attenuated in both the CL1-3 and PE089 cells (Figure 4B). Further, the influence of silencing AKR1B10 on migration and invasion abilities were also determined. The CL1-3 or PE089 cells that were AKR1B10-silenced elicited significantly reduced migration and invasion abilities when compared to the sc-transfected cells (Figure 4C).

### 3.6. Silencing AKR1B10 Greatly Ameliorates Cellular Sensitivity to Chemotherapeutics

Because the AKR1B10 expression was pertinently associated with oncogenic properties (Figure 4), it was postulated that the cellular sensitivity of NSCLC might be modulated by AKR1B10. The taxol- and doxorubicin-treated cells that elicited significantly upregulated AKR1B10 were examined for alterations in cellular sensitivity. Our data demonstrate that the colony-forming ability of the AKR1B10-silenced cells was significantly lowered in the cells treated with taxol and doxorubicin (Figure 5A,B). Of note, when cells were treated with cisplatin or gemcitabine, the AKR1B10 silencing also led to significantly decreased number of colonies formed (Figure 5C,D), implicating a general association of AKR1B10 to mediating cellular sensitivity to chemotherapeutics.

### 3.7. AKR1B10 Expression Modulates ERK Activation

Cyclins A, D, and E have long been associated with lung cancer progression [31,32,33]; nonetheless, very little regarding its correlation to chemoresistance regulated by AKR1B10 has been uncovered. Next, the efficacy of AKR1B10 overexpression in CL1-0 and CL1-3 cells was verified by immunoblotting and activity analysis (Figure 6A,B). Figure 6C demonstrates that AKR1B10 significantly increased the growth rates in both CL1-0 and CL1-3 cells as compared to the control transfected counterparts (Con). As shown in Figure 6D, ERK activation, as manifested by phosphorylated ERK, was markedly enhanced upon increasing amounts of AKR1B10 transiently overexpressed in both CL1-0 and CL1-3 cells. ERK was shown to be an important pathway by which AKR1B10 regulates NSCLC progression. Taken together, our results provide a novel oncogenic axis composed of AKR1B10 that can efficiently drive NSCLC proliferation, migration, invasion, and drug resistance after treatment with sublethal doses of chemotherapeutic agents (Figure 6E).

## 4. Discussion

Acquired drug resistance to chemotherapeutics has been extensively investigated to alleviate patients’ tumor burdens in cancer research [34]. Most studies to date have focused on chemotherapeutic drugs in regard to acquired chemoresistance through continuous drug exposure with a single high-dose treatment or incremental concentrations [34,35], while very few reports have investigated the chemoresistance acquired after the cessation of treatments with sublethal doses. In this study, we delved into the repercussions of administering sublethal chemotherapies on the malignancy of lung cancer. Our investigation unveiled a pivotal role played by AKR1B10 in the regulation of cell proliferation, metastasis, and chemoresistance.

AKR1B10 expression is upregulated and has been proposed as a prognostic biomarker in breast and pancreatic cancer types [36,37]. This is in line with our observations of the clinicopathological analysis using datasets from the GPL570 platform, TCGA, and Kaplan–Meier (Figure 3) that advocated the significant association of the AKR1B10 expression with the disease progression of lung cancer.

Recently, AKR1B10 was also implicated in promoting the progression of primary lung cancer and conferring resistance to cisplatin [25,38], anthracycline in breast cancer [39], and doxorubicin in gastric cancer [40]. Nevertheless, the underlying mechanism by which AKR1B10 contributes to chemoresistance acquisition remains largely elusive. In the present study, AKR1B10 stood out as the novel gene most significantly upregulated in taxol- and doxorubicin-treated NSCLC cells at sublethal doses (Figure 3). Silencing AKR1B10 demonstrated significantly reduced resistance to all four chemodrugs investigated (Figure 5). In fact, sublethal taxol, doxorubicin, cisplatin, and gemcitabine treatments that resulted in elevated cellular migration, invasion, and metastasis in NSCLC (Figure 2, Table 1) echoed an early finding in breast cancer where sublethal doxorubicin promoted migration and invasion via the regulation of Src family kinases [15]. Consistent with our finding in Figure 4B, which showed the cellular level of AKR1B10 to be critical for cyclin expressions and cell cycle regulation, the cyclin D1 was also reported to be regulated by AKR1B10 in breast cancer [41]. Further, the downregulation of AKR1B10 deactivates ERK activation in bladder cancer via CBX7 [42]. Our data revealed that ERK activation, as evidenced by elevated ERK phosphorylation in ARK1B10, caused an overexpression that led to augmented cellular growth (Figure 6).

There have been some reports on the upstream mechanisms of AKR1B10. AKR1B10 is upregulated by EGF and insulin through AP-1 signaling and is associated with the development of liver cancer [43]. Furthermore, the regulation of miR-137/AKR1B10 by long non-coding RNA (lncRNA) 1700020I14Rik affects inflammatory responses and hepatocellular damage [44], indicating that we can reduce the expression of AKR1B10 by inhibiting EGFR and insulin receptor signaling pathways or by increasing miR-137. Of note, the PE089 used in our study harbors the epidermal growth factor receptor (EGFR) exon 19 deletion that elicits resistance to the EGFR tyrosine kinase inhibitor (EGFR TKI) [45], implying its potential to serve as a dual chemoresistance study model of NSCLC.

Hypoxia, a reduction in oxygen content at the cellular or tissue level, is a characteristic of most tumors. In a hypoxic environment, the proliferation rate of cancer cells may slow down, and their invasion and migration rates may increase [46]. In this study, we examined the effects of sublethal chemotherapy drugs on lung cancer cells. Just like hypoxia on cancer cells, sublethal doses of drugs produced many physiological changes in cancer cells to adapt to this situation. It also shows that undertreatment not only does not significantly help but also may make patients worse off. In our study, the ABR1B10 expression increased after the sublethal dose treatment; accordingly, knocking down this gene’s expression may help improve chemotherapy’s efficacy and may provide a novel approach to improving treatment outcomes in cancer patients.

## 5. Conclusions

Sublethal exposition to chemotherapy rendered the NSCLC cells refractory after ceasing treatments, during which the AKR1B10 expression was markedly elevated. Silencing or overexpressing AKR1B10 significantly amended the cellular proliferation, migration and invasion. Repercussions resulted from chemoresistance acquisition such as Erk activation and the expressions of cyclins, and the regulation of the cell cycle could be pertinently modulated by manipulating the cellular AKR1B10 expression, underscoring its functional role in the progression of NSCLC. In short, our present findings supported the pivotal role of AKR1B10 in mediating the temporal evolution of chemotherapy resistance in NSCLC.

## Figures and Tables

**Figure 1 cancers-16-02428-f001:**
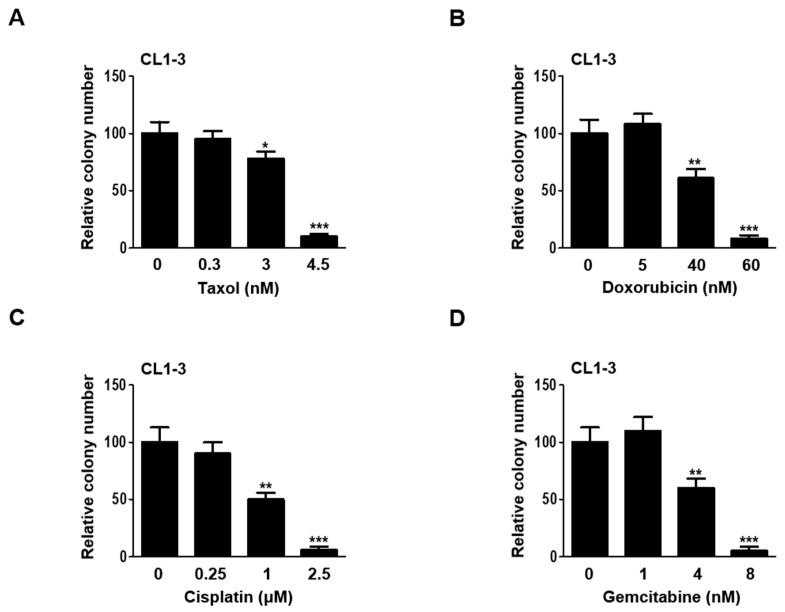
Cytotoxicity of the four anticancer drugs on CL1-3 cells. The CL1-3 cells were seeded into 6-well plates for colony formation assay. The cells were treated with four anticancer drugs: (**A**) taxol, (**B**) doxorubicin, (**C**) cisplatin, and (**D**) gemcitabine, at different concentrations. * *p* < 0.05, ** *p* < 0.01, *** *p* < 0.001.

**Figure 2 cancers-16-02428-f002:**
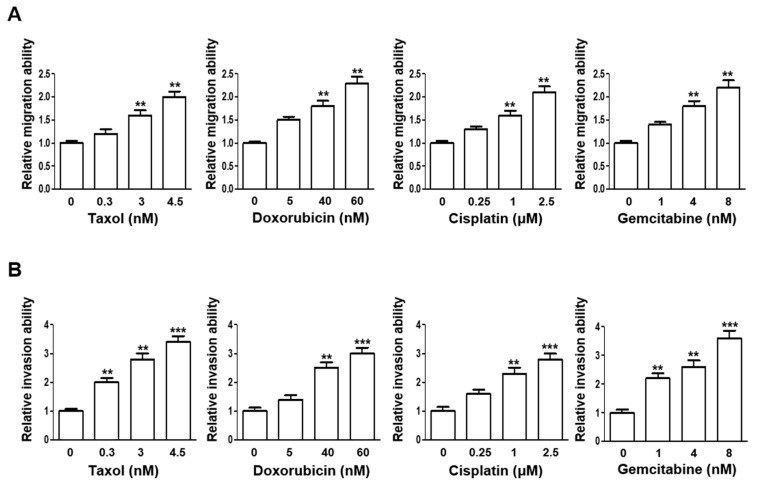
Increase in migration and invasion abilities after recovery of sublethal low dosages chemotherapy drugs treatment. CL1-3 cells were exposed to taxol, doxorubicin, cisplatin, and gemcitabine at the indicated doses for 3 days, then passaged 2 times, cultured in drug-free medium, and used for (**A**) migration and (**B**) invasion ability assays. ** *p* < 0.01, *** *p* < 0.001.

**Figure 3 cancers-16-02428-f003:**
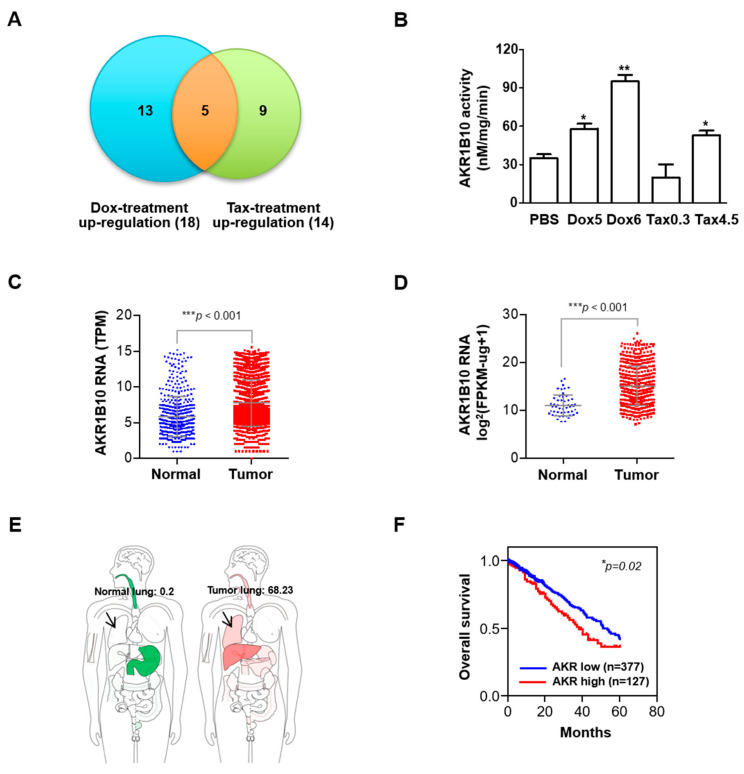
AKR1B10 expression and activity are induced after sublethal low doses of the chemotherapy drugs treatment. (**A**) Venn cluster analysis was used to analyze the microarray experiments that utilized mRNA extracted from the CL1-3 cells treated with sublethal doses of chemotherapeutics taxol (Tax) and doxorubicin (Dox). AKR1B10 stood out as one of the five top differentially expressed genes. (**B**) Detection of AKR1B10 activity in the CL1-3 cells recovered after two passages post-treatment with doxorubicin (Dox 5nM, Dox 60 nM) and taxol (Tax 0.5 nM, Tax 4.5 nM). Scatter plot of AKR1B10 mRNA expression in lung cancer patients from (**C**) the GPL570 platform dataset or (**D**) the GDC TCGA LUAD dataset. (**E**) Body map of the ARK1B10 transcript levels in human normal (left, green) or tumor tissues (right, red). Specific levels in lungs were as indicated (arrows; normal lung: 0.2; tumor lung: 68.23). The data were obtained from GEPIA2. (**F**) Kaplan–Meier analysis for the correlation between the AKR1B10 (AKR) expression and overall survival of lung adenocarcinoma patients (*n* = 504). * *p* < 0.05, ** *p* < 0.01, *** *p* < 0.001.

**Figure 4 cancers-16-02428-f004:**
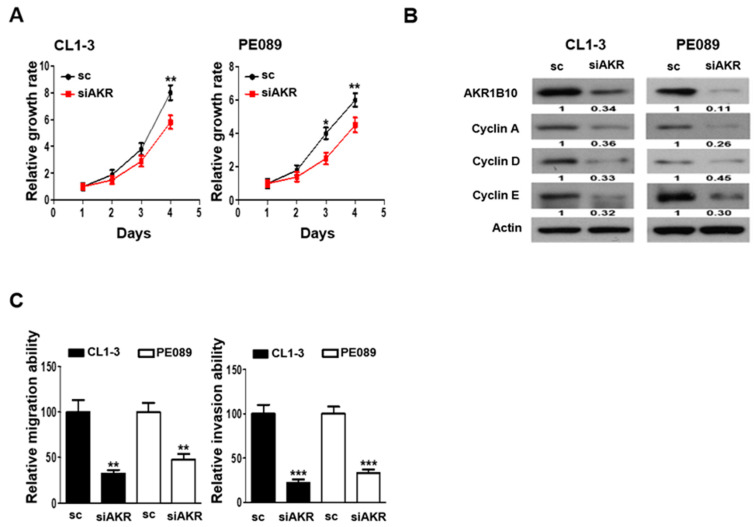
AKR1B10 silencing leads to reduced cellular proliferation, lower cyclin expressions, migration and invasion. (**A**) Cellular proliferation of the AKR1B10-silenced (siAKR) CL1-3 and PE089 cells were determined by comparing them to the scramble control (sc) cells using MTT assay. (**B**) Western blotting was employed to examine the changes in the protein expression of AKR1B10, cyclin A, cyclin D1, and cyclin E in the CL1-3 or PE089 cells transfected with sc or siAKR. (**C**) The migration and invasion abilities of the CL1-3 or PE089 cells transfected with sc or siAKR were determined. * *p* < 0.05, ** *p* < 0.01, *** *p* < 0.001. The original Western blot figures can be found in Appendix A.

**Figure 5 cancers-16-02428-f005:**
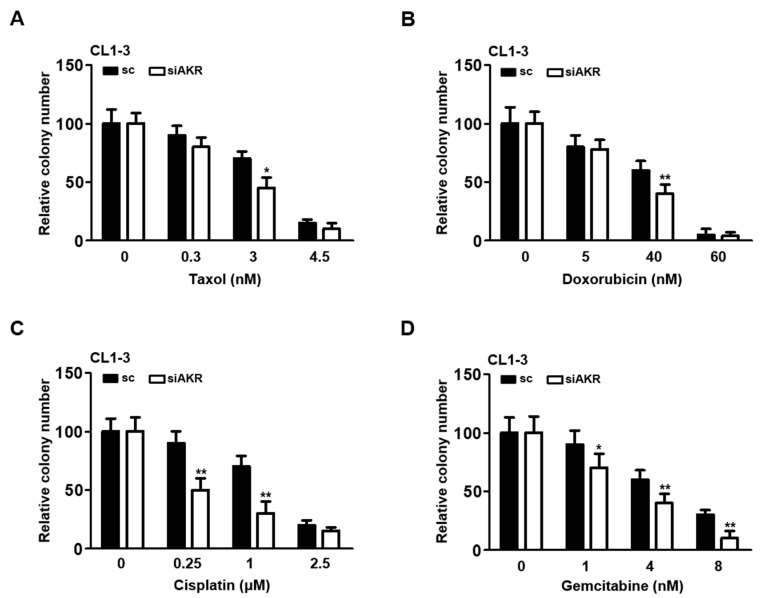
AKR1B10 silencing improves cellular sensitivity to chemotherapeutic treatments. Colony formation assays were conducted using scramble control or AKR1B10-silenced CL1-3 cells that were treated with various doses of (**A**) taxol, (**B**) doxorubicin, (**C**) cisplatin, and (**D**) gemcitabine. * *p* < 0.05; ** *p* < 0.01.

**Figure 6 cancers-16-02428-f006:**
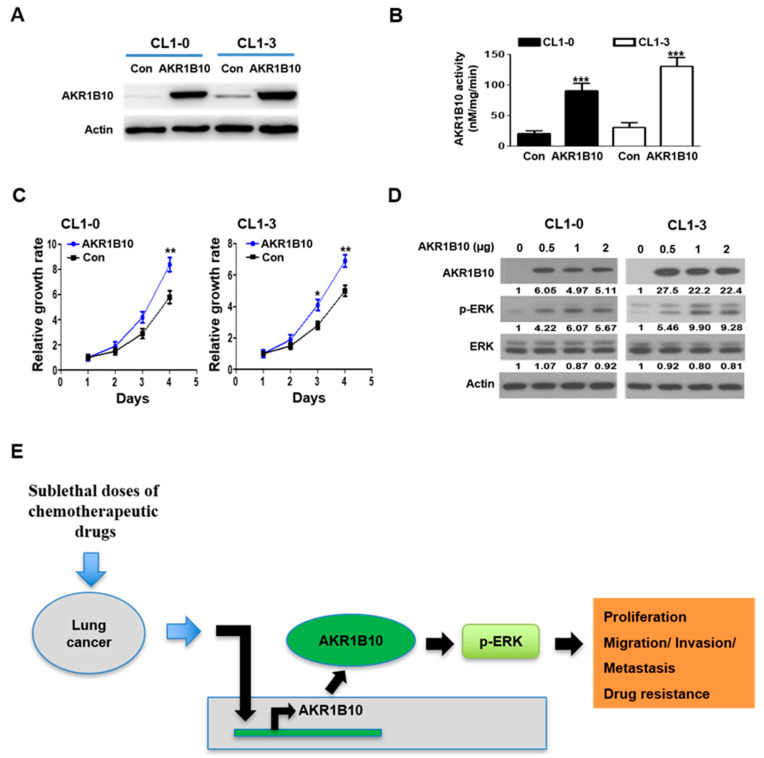
AKR1B10 promotes cell progression via ERK pathway activation. (**A**,**B**) Immunoblotting and activity were used to assess the AKR1B10 protein and activity expression in CL1-0 and CL1-3 cells that had been transfected with the control plasmid (Con) or AKR1B10 plasmid. (**C**) Growth of CL1-0 and CL1-3 cells that were transfected with the Con or AKR1B10 plasmid were determined by MTT assays. (**D**) Western blot analysis of the total ERK and phospho-ERK in CL1-0 and CL1-3 cells. (**E**) A schematic diagram that depicts a novel signaling mechanism of sublethal chemotherapy doses in non-small-cell lung cancer. * *p* < 0.05, ** *p* < 0.01, *** *p* < 0.001. The original Western blot figures can be found in Appendix A.

**Table 1 cancers-16-02428-t001:** In vivo metastasis assay. CL1-3 lung cancer cells received sublethal chemotherapy pre-treatment, followed by two passages of culturing. These treated cells were then injected into the mice via the tail vein. After 6 weeks, the mice were examined for lung metastases.

	Number of Lung Nodules (Mean ± SD)	*p* Value	*n*
PBS	2.9 ± 1.5		14
Tax 0.3	3.9 ± 3.4	0.309	11
Tax 4.5	6.8 ± 2.9	0.000 **	11
Dox 5	4.2 ± 1.2	0.021 *	12
Dox 60	6.0 ± 1.7	0.000 **	13
Cis 0.25	5.0 ± 2.2	0.006 **	13
Cis 2.5	5.8 ± 2.6	0.001 **	11
Gem 1	5.6 ± 1.9	0.000 **	12
Gem 8	6.5 ± 3.8	0.003 **	13

Abbreviations: taxol (Tax in nM); doxorubicin (Dox in nM); cisplatin (Cis in μM); gemcitabine (Gem in nM). * *p* < 0.05; ** *p* < 0.01.

## Data Availability

The data presented in this study are available on request from the corresponding authors.

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
