# Peer review of "The Role of AKR1B10 in Lung Cancer Malignancy Induced by Sublethal Doses of Chemotherapeutic Drugs"

_cancers, 2024, doi:10.3390/cancers16132428_

Round 1
Reviewer 1 Report
Comments and Suggestions for Authors
1) The authors presented their findings in the abstract qualitatively, but it is important to present their numerical results.
2) I strongly recommend the authors to draw a graphical abstract at the end of the introduction.
3) The authors are required to describe more about the AKR1B10 in the introduction since a lot of readers may not know about it.
4) Why these two cell lines (CL1-3 and PE089) was used while A549 is an abundant cell line among NSCLC.
5) It is required to draw a table representing the two cell lines treated with x nM chemotherapy agent. Moreover, the cell lines, which are treated with two anti-cancer drugs simultaneously must be mentioned. Is the concentration of chemotherapy drugs “same” for AKR1B10 transfected and non-transfected? This part is very confusing.
6) Why only the CL1-3 cell line was used in the 2.5, 2.7, and 2.9 sections.
7) Why the chemotherapy agents in Figure 1 were presented in a different mode? ( nM and ng/mL) the same unit is better for comparison.
8) Why do relative migration ability and invasion ability increase along with increasing chemotherapy drug concentrations? (figure 2)
9) Did the authors use a combination of the chemotherapy drugs? (to investigate the synergistic effects)
10) Why did the authors select these drugs since the mechanisms of them are different? (e.g. taxol affects on cell cytoskeleton, doxorubicin produces ROS, and cisplatin damages DNA)
11) How the results of cellular studies can be generalized to clinical outcomes since cancer is a very heterogeneous disease?
Reviewer 2 Report
Comments and Suggestions for Authors
The reviewed manuscript is quite interesting from a scientific point of view and very useful for clinical practice. The aim of the research is evaluation of the impact of chemotherapeutical drugs in suboptimal (low) doses on progression of malignancies and on chemoresistance development.
An important role of AKR1B10 (a member of the Aldo-keto reductase family), after its exposure to suboptimal (low) doses of chemotherapeutic drugs, has been found out in proliferation, migration, and activation of non-small cell lung cancer (NSCLC). In the last decade AKR1B10 has attracted attention of various specialists in oncology as a homeostasis component which plays an ambivalent role in malignancies. Several published works demonstrate its usage as a prognostic factor for various cancer diseases. Others also contain data on its stimulation of progression and metastasis of malignant tumors. On the contrary, there are other data on its inhibition of tumor growth.
The data obtained in this research could probably play a pivotal role in understanding the chemoresistance mechanisms, as well as in further development of novel approaches to creation of medicines for changing the AKR1B10 expression.
The reviewed research is original and advances the current knowledge.
The authors have performed a detailed research and studied the main patterns of behavior of cultured NSCLC cells (2 models) under the impact of widely used chemotherapeutic drugs in various doses. The obtained in vivo evidence proved the impact of AKR1B10 expression level in tumor cells exposed to suboptimal doses of cytostatic agents on the ability of these cells to induce tumor nodes in lungs.
Relevant modern experimental methods have been used in vitro and in vivo. Their description is clear and detailed and, it is quite valid for reproduction of the results by other researchers.
The results are well structured and clearly presented. The illustrations support the results and help to understand them easily. The statistical analysis of the numerical data has been done according to modern guidelines for statistics processing.
The obtained data are discussed in comparison with works on the similar subjects. The conclusions are based on critical evaluation of the results and demonstrate the authors’ understanding of the future direction of research which may be very important for the development of antitumor therapy.
The manuscript is written in good literary English. Minor correction of the language may be done by the specialist in English.
Still some points of the manuscript should be corrected.
The term “sublethal doses of chemotherapeutic drugs” in the context of this manuscript is not fully applicable. The concept “sublethal” refers to toxicology and is applied to animals which do not die after being injected this dose but die after the treatment with larger lethal doses. The term “suboptimal doses” is more correct for description of the cultured cells because it demonstrates lack of the necessary cytotoxic efficacy of the drug used in this dose in vitro. Unfortunately, sometimes the term “sublethal doses” is used in some other publications.
The usage of terms “metastasis” and “the development of metastases” is not correct when they are applied to tumors induced by intravenous infusion of tumor cells to animals. Metastasis in oncology means secondary tumor nodes of the same histological structure as the primary spontaneous or somehow induced node. Multiple lung tumors after infusion of tumor cells into murine blood stream described in this manuscript must be termed as primary nodes, and not metastases.
The work fits the scope of the Cancers journal and is undoubtedly interesting for various specialists: clinical and experimental oncologists, biochemists, molecular biologists, genetics, and others.
The manuscript meets the standards of publication ethics. It may be accepted for publication after minor editing according to the reviewer’s comments.
Comments on the Quality of English LanguageEnglish language fine. Some stylistic incorrectness should be eliminated by the specialist.
Reviewer 3 Report
Comments and Suggestions for Authors
Dear authors,
After reviewing this interesting study, I have several concerns and suggestion regarding to this study:
1. I would like to know the reason of choosing CL1-3 and PE089 as your model cell line rather than more common-used cell lines such as NCI-H1975.
2. I would like to know the reason of utilizing doxorubicin in this study. Based on the practical guideline, cytotoxic agents used in NSCLC treatment include carboplatin, paclitaxel, gemcitabine, pemetrexed, etoposide, and vinorelbine. Doxorubicin is not a common-used chemotherapeutic agent in NSCLC treatment.
3. I think the rationale of investigating AKR1B10 in NSCLC is not quite sufficient. I would suggest the authors including the literature about AKR1B10-involved brain metastasis and chemoresistance (such as 10.1186/s12967-023-04403-0 and 10.1016/j.actbio.2019.04.053) and that can reinforce the rationale of investigating the role of AKR1B10 in chemotherapy-induced resistance.
4. Do the authors develop chemotherapy-resistant subclone of the cell model? I think the expression of AKB1B10 in such subclone and the functional change while manipulating expression of AKB1B10 in such subclone can potentiate the crucial role of AKB1B10in chemotherapy-induced chemoresistance.
5. From the previous suggestion, I think the experimental designation of Figures 4 & 5 can be improve. If the authors can repeat the experiment in Figures 4 & 5 and replacing the model cells from treatment-naive cell to treatment-experienced cells, the persuasiveness of this experiment can be largely improved.
6. This study focused on the downstream mechanism of chemotherapy-induced AKB1B10 up-regulation but discussed less about the potential upstream regulation. I suggest the authors discuss about such issues.
7. Do the authors analyze the distribution of AKB1B10 in your model cell population? Because your study is under sub-lethal dosage, which may generate a force for selecting AKB1B10-high subclone due to their chemoresistant nature.
Reviewer 4 Report
Comments and Suggestions for Authors
In their current manuscript, Jang et al. conducted research on the role of Aldo-keto reductase family 1 member B10 (AKR1B10) in non-small cell lung cancer (NSCLC) metastasis during chemoresistance. The authors highlighted the overexpression of AKR1B10 in NSCLC cell lines at sub-lethal doses of four drugs and discussed its role in the ERK pathway. The research was conducted thoroughly, and the manuscript is well-written. I have a few minor points or comments mentioned below:
1. Many abbreviations are not properly described. I suggest that the full forms be provided the first time they are used. Examples include RIPA buffer, GPL570, GDC TCGA LUAD, etc.
2. Section 2.7: The authors mentioned the reagents for AKR1B10 but did not detail the exact method for assaying AKR1B10 activity.
3. Line 154: Please correct the symbol "20uM" to "20 micromolar."
4. I suggest using the same unit for all drug concentrations. I noticed that the concentrations for three drugs (taxol, cisplatin, and doxorubicin) are given in micromolar or nanomolar, but for gemcitabine, it is in ng/ml.
5. Section 3.4: Only the results for two drugs (doxorubicin-treated and taxol-treated) are mentioned. What about the other two drugs?
6. Figure 3B: Please clarify the meaning of "D5," "D60," "T03," and "T4.5" in the figure legend.
7. Section 3.6: The authors mentioned the silencing results for the CL1-3 cell line. Why did they not mention another cell line, PE089?
8. Line 297: “Figure 6C demonstrates AKR1B10 resulted in significantly...” What does "resulted" mean in this context?
9. Line 358: "sturdy" should be corrected to "study."
Round 2
Reviewer 1 Report
Comments and Suggestions for Authors
The authors performed my corrections. Only two items have been left: numerical results in the abstract and graphical abstract at the end of the introduction.
Reviewer 3 Report
Comments and Suggestions for Authors
Dear authors,
Thank you for your kindly response to my concern. I have no further concerns regarding this study except one suggestion. The choose of doxorubicin in this study would be a weak point because doxorubicin is not a conventional-used agents in NSCLC and SCLC. That is, the mechanism regarding to doxorubicin-induced AKR1B10 upregulation cannot refer to clinical phenomenon. If possible, I would suggest the authors weakening the description of doxorubicin in this study. Also, the suggestion regarding chemoresistant subclone could be enlisted in the prospective which can be a following topic.
